# Dual Thermo- and Photo-Responsive Micelles Based on Azobenzene-Containing Random Copolymer

**DOI:** 10.3390/ma15010002

**Published:** 2021-12-21

**Authors:** Chuan Yan, Liqin Yang, Xiangquan Mo, Keying Chen, Weiya Niu, Zhiju Zhao, Guanghua Li

**Affiliations:** 1Guangxi Key Laboratory of Petrochemical Resource Processing & Process Intensification Technology, School of Chemistry and Chemical Engineering, Guangxi University, Nanning 530004, China; 19113010010@fudan.edu.cn (C.Y.); moxiangquannl@163.com (X.M.); simaqingyue@sina.cn (K.C.); 2Hebei Functional Polymer Materials R&D and Engineering Application Technology Innovation Center, College of Chemistry & Chemical Engineering, Xingtai University, Xingtai 050041, China; Yangli2005@163.com (L.Y.); niuweiya0021@163.com (W.N.)

**Keywords:** methacrylamido-azobenzene, 2-hydroxyethylacrylate, amphiphilic random copolymer, micelles, Nile red, photo-/thermo-responsiveness

## Abstract

Amphiphilic random copolymer poly(methacrylamido-azobenzene)-*ran*-poly(2-hydroxyethylacrylate) (PMAAAB-*ran*-PHEA) was synthesized via hydrolysis of poly(methacrylamido-azobenzene)-*ran*-poly[2-((2′-tetrahydropyranyl)oxy)ethylacrylate] (PMAAAB-*ran*-P(THP-HEA)), which was prepared by conventional radical polymerization. PMAAAB-*ran*-PHEA micelles were then prepared via dialysis method against water with DMF as solvent. The structure, morphology, size, and low critical solution temperature (LCST) of PMAAAB-*ran*-PHEA and its micelles were determined by ^1^H-NMR, GPC, TEM, and DLS. The thermo- and photo-responsive behaviors of the resulting polymer micelles were investigated with Nile red as a fluorescence probe. The results showed that PMAAAB-*ran*-PHEA micelles were porous or bowl-shaped and its size was 135–150 nm, and its LCST was 55 °C when *F*_MAAAB_ of the random copolymer was 0.5351; the hydrophobicity of the micellar core was changed reversibly under the irradiation of UV light and visible light without release of Nile red or disruption of micelles; the size and solubilization capacity of the micelles were dependent on temperature, and Nile red would migrate for many times between the water phase and the micelles, and finally increasingly accumulated during the repeated heating and cooling processes.

## 1. Introduction

Stimuli-responsive micelles are the aggregates in solution which consist of a core formed by the hydrophobic blocks and a shell composed by hydrophilic blocks, featuring some special functional groups which can respond to various chemical and physical stimuli such as pH, temperature, light, ions and redox [1,2,3,4,5,6]. The potential applications of stimuli-responsive micelles in controlled release, drug delivery, biosensors, catalyst system, and nano-reactors [7,8,9,10,11] and are attributed to the polymeric micelles changes induced by the responsive moieties in morphology, diameter, polarity and structure under the external stimuli.

Most of the stimuli-responsive micelles reported so far have been focused on the preparation and application of the diblock or triblock copolymers [12,13,14,15]. Nevertheless, since Liu et al. [16] reported on the preparation of bowl-shaped aggregates from the self-assembly of an amphiphilic random copolymer of poly(styrene-*co*-methacrylic acid) [P(St-co-MAA)] in 2005, the polymeric micelles composed by the amphiphilic random copolymers have attracted significant attention. Dey et al. [17] used PEG-based random copolymer micelles as drug carriers and investigated the effect of hydrophobe content on drug solubilization and cytotoxicity. Terashima et al. [18] synthesized and characterized a size-controlled and thermo-responsive micelles composed of amphiphilic random copolymers bearing hydrophilic poly(ethylene glycol) (PEG) and hydrophobic alkyl pendants.

To the best of our knowledge, it is the single stimulus that has been most extensively studied so far. In contrast to other stimuli (i.e., temperature, pH), light is more special, because it makes temporal and spatial control accessible simultaneously. It is also known to us that light-responsive polymeric micelles based on the photoisomerization of azobenzene have been widely studied in recent years. Wang et al. [19,20] reported that the colloidal particles were disaggregated upon UV irradiation. Geng et al. [21] demonstrated that the release of rhodamine B (RhB) and doxorubicin (DOX) from a self-assembly formed from a cationic azobenzene derivative could be triggered by UV irradiation. Stenzel et al. [22] observed the hydrophobic dye Nile red encapsulated in micelles was taken up by A375 human melanoma cells with UV irradiation. 

In recent years, it is the need of more smart and multiple nano-structured materials in medical and biotechnology applications that facilitates the rapid progression of dual responsive polymeric micelles, especially by using temperature and light as external stimuli [2]. As mentioned above, the reason inducing the change of molecule with azo-containing moieties in shape and polarity under UV irradiation is the existence of the isomerization of azobenzenemoieties. On the other hand, it is well known that poly(*N*-isopropylacrylamide) (PNIPAM) has a significant thermo-responsiveness because of its lower critical solution temperature (LCST) in water, and an obvious phase transition from a soluble to an insoluble state took place by tuning the ambient temperature below or above its LCST. In 2010, Yu et al. [23] synthesized a novel amphiphilicdiblock copolymer composed of a hydrophilic PEG block and a hydrophobic PAZO-*co*-PNIPAM block, and the polymeric micelles showed dual responsiveness to heat and light. Recently, Pinol et al. [24] investigated the thermo- and photo-responsive properties of the miktoarm star polymeric micelles with Nile red as a fluorescence probe, and then they observed that Nile red was released from the micellar cores under the irradiation of UV light because of the *trans*-to-*cis* isomerisation of the azobenzene moieties. However, temperature-induced collapse of the micelles was not suitable for Nile red release.

Polymeric micelles with multi-responsiveness developed rapidly and obtained extensive attraction from researchers in recent years, and one of the most important reasons for this was that micelles with multiple stimuli-responsiveness are more likely to achieve more functionalities and be modulated through more parameters and bring some all-new properties [25]. Thus, in this research, we reported dual thermo- and photo-responsive micelles through the self-assembly of the amphiphilic random copolymer of poly(methacrylamido-azobenzene)-*ran*-poly(2-hydroxyethylacrylate) (PMAAAB-*ran*-PHEA). The size of the micelles changed obviously with ambient temperature, and the controlled release of encapsulated Nile red in micelles were also observed during heating and cooling processes. Moreover, the reversible change of hydrophobicity of the micellar cores occurred under the irradiation of UV light and visible light without release of Nile red or disruption of micelles.

## 2. Methods

### 2.1. Materials

Methacrylamido-azobenzene (MAAAB) and 2-[(2′-tetrahydropyranyl)oxy]ethylacrylate (THP-HEA) monomer were prepared in our laboratory according to previous reports [26,27]. Absolute diethyl ether (AR, XilongScientific, Shantou, China), ethyl acetate (AR, XilongScientific, Shantou, China), petroleum ether (AR, XilongScientific, Shantou, China), concentrated hydrochloric acid (HCl, AR, Aladdin, Shanghai, China), methanol (AR, XilongScientific, Shantou, China), Nile red (NR, 95.0%, Aladdin, Shanghai, China), *N,N*-dimethyl formamide (DMF, 95%, XilongScientific, Shantou, China) were used without further purification. Tetrahydrofuran (THF, 99.8%, Sinopharm Chemical Reagent, Shanghai, China) was refluxed with sodium and distilled. 2,2′-Azobis (isobutyronitrile) (AIBN, 98.0%, Aladdin) was recrystallized from absolute ethanol and stored in refrigerator.

### 2.2. Synthesis of PMAAAB-ran-P(THP-HEA)

PMAAAB-*ran*-P(THP-HEA) copolymers were synthesized by following the procedures as shown in Figure 1.

Methacrylamido-azobenzene (MAAAB), 2-[(2′-tetrahydropyranyl)oxy]ethyl acrylate (THP-HEA) with a certain proportion, AIBN(4.1 mg, 0.025 mmol) and 80.0 mL THF were added to a 150 mL round flask, and were stirred until a homogeneous solution was formed. After being flushed by nitrogen for 40 min, the flask was sealed and placed in an oil bath at 60 °C for some time. The conversion was controlled below 10% by adjusting the reaction time. After the reaction was over, the solvent of mixture was partially evaporated and the resulting polymer solution was purified by dissolving it in THF and precipitated in a mixture solution(methanol/water = 4:1 (*v*/*v*) for 3 cycles. After being dried in high vacuum at 60 °C, a series of PMAAAB-*ran*-P(THP-HEA)random copolymers were obtained.

### 2.3. Synthesis of PMAAAB-ran-PHEA

PMAAAB-*ran*-PHEAcopolymers were synthesized by the following procedures as shown in Figure 1.

Random copolymers (0.20 g) with different monomer proportions of MAAAB and THP-HEA, HCl (2.0 mL, 89 µmol) in 18.0 mL water, and 40.0 mL THF were added to a 100 mL round flask to form a homogeneous solution. The solution was stirred at 65 °C for 8 h and then cooled to room temperature. The resulting copolymer solution was precipitated into cold (−10 °C) *n*-hexane (300 mL) for 2 cycles to yield 0.17 g (85%) of PMAAAB-*ran*-PHEA.

### 2.4. Preparation of PMAAAB-ran-PHEA Micelles 

0.01 g of PMAAAB-*ran*-PHEA random copolymer was dissolved in 40 mL DMF, and stirred until a homogeneous solution was formed. Then the resulting solution was dialyzed against deionized water for 4 days to form micelles, using a regenerated cellulose membrane tubing (molecular weight cut off 1000). To remove the solvent DMF completely, the water was exchanged every 6 h regularly in the formation process of micelles. Finally, polymer micellar solution was obtained by dilution of the deionized water.

### 2.5. Encapsulation of Nile Red

The micelles with encapsulated Nile red were obtained as follows: 2.0 mL of 0.1 mgL^−1^ solution of Nile red in THF was added to a 250 mL conical bottle and the THF was evaporated at 60 °C. 100 mL of 0.1 mgL^−1^ polymer micellar solution was added to the bottle, and then the bottle was sealed and heated at 60 °C for 6 h to equilibrate the Nile red and the micelles, and subsequently allowed them to cool to room temperature.

### 2.6. Characterization

^1^H-NMR spectrum of the copolymers for PMAAAB-*ran*-P(THP-HEA) and PMAAAB-*ran*-PHEA was recorded in CD_3_COCD_3_ solvent using a Bruker ADVANCE III HD 600 MHz spectrometer (Bruker Group, Zurich, Switzerland). The molecular weight and its distribution of the random copolymers were determined by GPC equipped with a Waters 1515 HPLC pump, Waters 2414 refractive index detector, and Waters styragel HR series (HR1, 3, 4) (eluent: THF, 35 °C, flow rate: 1 mL/min). The column system was calibrated by a polystyrene standard. The size and its distribution of random copolymeric micelles were determined in a LS Instruments LSI-3DLS (Swiss LS Instrument Co., Fribourg, Switzerland) using a He-Ne laser with a 632.8 nm wavelength, a detector angle of 90 at 25 °C. The morphology of the random copolymeric micelles with the concentration of 0.02 gL^−1^ was observed by TEM in a JEM-1200EX18 (Japan Electronics Co., Ltd., Tokyo, Japan) instrument operated at 80 kV. UV-vis spectrum of the micellar solution was recorded by using a PERSEE UV-1810SPC UV-vis spectrophotometer (Beijing General Instrument Co., Ltd., Bejing, China) using a UV lamp (365 nm, 200 μWcm^−2^) as a light source. The fluorescence spectrum of the micellar solution was measured at an excitation wavelength of 587 nm using a CARY ECLIPSE spectrofluorophotometer (Agilent Technologies Inc., Palo Alto, CA, USA) at 25 °C.

## 3. Results and Discussion

### 3.1. Synthesis of PMAAAB-ran-P(THP-HEA)

The random copolymer PMAAAB-*ran*-P(THP-HEA) was synthesized by conventional radical polymerization (CRP) using MAAAB, THP-HEA as monomers, AIBN as initiator. The conversion of random copolymers composed of different proportions of MAAAB and THP-HEA was controlled below 10%, and the synthetic data, copolymer composition (*F*_MAAAB_)and number-average molecular weight ($\bar{M}$_n_) of the random copolymers are shown in Table 1. Figure 1 shows the GPC trace of PMAAAB-*ran*-P(THP-HEA) (*F*_MAAAB_ = 0.5351), and the number-average molecular weight ($\bar{M}$_n_) and its distribution ($\bar{M}$_w_/$\bar{M}$_n_) of PMAAAB-*ran*-P(THP-HEA) were 11,990 and 1.50, respectively. From Table 1, an obvious increase of *M*_n_ is observed with the proportion of *n*_MAAAB_:*n*_THP-HEA_ gradually increasing from 2:8 to 5:5, while copolymers’ *M*_n_ begins to decrease when the proportion of *n*_MAAAB_:*n*_THP-HEA_ further increased from 5:5 to 8:2. The representative ^1^H-NMR spectrum of PMAAAB-*ran*-P(THP-HEA) (*F*_MAAAB_ = 0.5351) is shown in Figure 2. The typical characteristic signals of protons in benzene ring (MAAAB) and acetal group(THP-HEA) were observed at about 7.2–8.0 ppm(l+m, j+k) and 4.6–3.2 ppm(i+h+g+f), respectively, which indicated the formation of PMAAAB-*ran*-P(THP-HEA), and the copolymer composition was calculated by using their peak area. Figure 3 shows the ^1^H-NMR spectra of copolymers with different compositions, a significant increase of peak area(D) at about 7.2–8.0 ppm of the typical characteristic signals of protons in MAAAB is observed with the proportion of *n*_MAAAB_:*n*_THP-HEA_ increasing from 2:8 to 8:2, in line with the gradual decrease of the THP-HEA characteristic peak area (E) at about 4.6–3.2 ppm simultaneously. 

### 3.2. Synthesis of PMAAAB-ran-PHEA

PMAAAB-*ran*-PHEA was synthesized via the hydrolysis of PMAAAB-*ran*-P(THP-HEA) with THF as solvent, HCl aqueous solution as catalyst. The representative ^1^H-NMR spectra of PMAAAB-*ran*-PHEA (*F*_MAAAB_ = 0.5351) and its precursor are shown in Figure 4. The typical characteristic signals of protons (l+m, j+k) in a benzene ring (MAAAB) were still retained, while the peaks on the pyran ring at about 4.6 ppm and 3.4 ppm were disappeared, which indicated the formation of PMAAAB-*ran*-PHEA.

### 3.3. Preparation of PMAAAB-ran-PHEA Micelles

The amphiphilic nature of PMAAAB-*ran*-PHEA random copolymer provides an opportunity to form micelles in water. The water soluble PHEA segments serve as the hydrophilic shell, and the PMAAAB segments form the hydrophobic core. Herein, PMAAAB-*ran*-PHEA random copolymeric micelles with different copolymer composition were prepared, and the hydrodynamic radius (*R*_h_) and morphology of micelles were determined by dynamic light scattering (DLS) and TEM, respectively. The size distribution of copolymer micelles by DLS is shown in Figure 5, and the results are listed in Table 2. The size of random copolymeric micelles was 135–150 nm, and the size of micelles firstly increased from 135.7 to 149.3 nm and then decreased to 141.0 nm, which match the variation of its number average molecular weight (*M*_n_) (which increased from 6240 to 11,990 firstly and then decreased to 6870). 

Figure 6 shows TEM images of the copolymer micelles S3, S4 and S5. The formation of porous- or bowl-shaped micelles are observed, as reported by Liu et al. [16], and the micelles size was close to DLS measurement. The formation of porous or bowl-shaped micelles may be related to the disordered arrangement of hydrophilic and hydrophobic segments in random copolymers and the preparation of micelles by dialysis against water.

### 3.4. Thermo-Responsive Behavior of Micelles

Due to the existence of the thermo-responsive moieties (PHEA segments) and photo-responsive moieties (PMAAAB segments), the random copolymeric micelles may exhibit a dual responsive behavior under external stimuli. The photo-responsiveness of amphiphilic random copolymeric micelles containing azobenzene side groups has been reported [19,20], and the change trend of PMAAAB-*ran*-PHEA micelles was the same as them; therefore, the thermo-responsiveness is only discussed here. We investigated the effect of the temperature on the *R*_h_ of PMAAAB-*ran*-PHEA micelles by DLS as shown in Figure 7, and the low critical solution temperature (LCST) of PMAAAB-*ran*-PHEA was determined from the graphical intersecting point. From Figure 7, an obvious decrease of *R*_h_ of the S3 sample was observed with the increase of external temperature, *R*_h_ was at about 109 nm and not changed when the external temperature was up to about 55 °C. According to the results, the LCST of S3 micelles was supported to be close to 55 °C, indicating that PMAAAB-*ran*-PHEA exhibited different solution properties at different ambient temperatures, which was hydrophilic at room temperature (approx. 25 °C) and became hydrophobic when the temperature was above the LCST. The LCST existence of random copolymers makes it possible to investigate the potential drug release and possible mechanism of micellar solution by changing the ambient temperature. 

### 3.5. Encapsulation of Nile Red and Its Photo- and Thermo-Induced Release

In order to study the photo- and thermo-induced release properties of random copolymeric micelles, a model hydrophobic dye molecule, Nile red, was encapsulated in it, serving as a fluorescent probe. Herein, the S3 micellar solution was selected (and has been used in a previous part) serving as host molecules loaded with Nile red. Figure 8 shows the TEM images of the micellar solution S3 and Nile red-loaded S3 (NR-S3). From the picture, the morphologies of S3 and NR-S3 were basically unchanged except for a little increase in size [28].

UV-visible spectra and fluorescence spectra of S3 and NR-S3 are shown in Figure 9, respectively. In Figure 9a, the Nile red-loaded NR-S3 solution not only still had a strong absorption band at about 360 nm, but also exhibited a characteristic absorption band of Nile red at 588 nm, compared with S3 micellar solution; in Figure 9b, a new characteristic fluorescence emission peak of S3 solution at about 645 nm was observed after being loaded with Nile red. According to the all above measurements, Nile red was successfully loaded.

Nile red is a special dye molecule, which does not exhibit any fluorescence when it is soluble in water, but the fluorescence intensity remarkably increased in the hydrophobic environment, for example, inside a hydrophobic micellar core [29]. In this case, the photo-induced Nile red release of S3 micellar solution was investigated first. Figure 10a shows that the fluorescence intensity at about 645 nm was obviously decreased when NR-S3 micellar solution was exposed to UV light, and the fluorescence intensity was not changed up to 70 min, indicating the random copolymeric azo-moiety reached a photoisomerization equilibrium. Figure 10b exhibited that the fluorescence intensity basically returned to the initial value, when in NR-S3 micellar solution after exposed to the UV light for 60 min was irradiated with visible light, and the fluorescence intensity was not changed up to 60 min. 

Figure 11a,b show the fluorescence spectra and maximum intensity value at photoisomeration equilibrium state under alternating irradiation of UV and visible light four times. As can be seen, the maximum intensity value was basically not changed at photoisomeration equilibrium state after repeating several times.

It is well known that the hydrophobic azobenzene polymer (PAZO) segments would become a little hydrophilic under the UV-light irradiation due to the photoisomerization of the azobenzene moieties [30] and the fluorescence intensity of Nile red becomes weaker in a less hydrophobic environment [31]. Therefore, the decrease in the fluorescence intensity of micellar solution was attributed to the more hydrophilic environment due to the PAZO photoisomerization under UV light irradiation. From Figure 10b, the fluorescence spectra returned to the initial value under the visible light irradiation, which indicated that there was no leakage of Nile red at the process of PAZO photoisomerization. On the other hand, from Figure 11, the fluorescence spectra and maximum fluorescence intensity value between *cis*-*trans* and *trans*-*cis* processes at the photoisomerization equilibrium point was not changed under alternating UV and visible light irradiation several times, indicating the micellar solution changed only in its internal hydrophobic/hydrophilic environment without any disruption of micellar structure under the irradiation of UV light. The photo-responsive micelles have a potential application in sensors or catalysts, which need the release of encapsulated substances to prevent pollution of the environment or could influence the chemical reaction system.

In order to investigate the influence of external temperature on micellar release properties, the fluorescence spectra of NR-S3 micellar solution was recorded at the heating or cooling process. Figure 12a shows that the fluorescence intensity of NR-S3 micellar solution at about 645 nm gradually decreased at the heating process from the room temperature, and the fluorescence spectra was not changed until the temperature was up to 50 °C (LCST = 55 °C). At the cooling process (Figure 12b), the fluorescence spectra of NR-S3 micellar solution kept increasing all the time, and the maximum intensity value was larger than before the former heating process at the identical temperature of 25 °C. From Figure 13a,b, an obvious increase of NR-S3 micellar solution maximum value (at 645 nm) at 25 °C was observed after alternating heating–cooling processes several times.

For the specific experimental results, refer to the influence of external temperature on NR-S3 micellar solution release properties, where a possible temperature-triggered release mechanism was proposed. As mentioned above, the random copolymeric micelles were composed of a hydrophilic shell of PHEA segments and a hydrophobic core of PMAAAB segments. At the heating process (Figure 12a), the PHEA segments became less hydrophilic, along with the shrinkage and size decrease of the micelles, leading to the migration of Nile red from the micelles to the water due to the decrease in the micellar solubilization capacity, and the decrease in the fluorescence intensity of the micellar solution, which also indicated that Nile red molecules encapsulated in micelles migrated from a hydrophobic environment to the water. At the cooling process (Figure 12b), the PHEA segments became more hydrophilic again, along with the expansion and size increase of the micelles, leading to more Nile red migrating from the water to the micelles due to the increase of the micellar solubilization capacity. Furthermore, the fluorescence intensity (Figure 13a,b) of micellar solution, which increased gradually under alternating heating–cooling procedures several times, was possibly attributed to the stronger solubilization capacity, indicating that the Nile red molecule can migrate many times between the water and the micelles by heating/cooling procedures. Furthermore, the Nile red molecule would increasingly accumulate from the aqueous environment to the micelles just by controlling the ambient temperature and repeating.

## 4. Conclusions

In summary, photo-/thermo-dual responsive micelles based on a random copolymer composed of a hydrophilic PHEA segments and hydrophobic PMAAAB segments were prepared. PMAAAB-*ran*-PHEA micelles were porous or bowl-shaped and their size was 135–150 nm, and their LCST was 55 °C when the *F*_MAAAB_ of random copolymer was 0.5351; the hydrophobicity of the micellar core was changed reversibly under the irradiation of UV light and visible light without release of Nile red or disruption of micelles; the size and solubilization capacity of the micelles were dependent on temperature, and Nile red would migrate many times between the water phase and the micelles by heating/cooling procedures and were increasingly accumulated from the aqueous environment to the micelles just by controlling the ambient temperature and repeating this a few times. The PMAAAB-*ran*-PHEA random copolymeric micelles may have a potential application in sensors, chemical reaction systems, and drug-controlled release.

## Data Availability

Not available.

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
