# Peer review of "Dual Thermo- and Photo-Responsive Micelles Based on Azobenzene-Containing Random Copolymer"

_materials, 2021, doi:10.3390/ma15010002_

Round 1

Reviewer 1 Report

In this submitted manuscript Yan et. al. prepared a series of amphiphilic random copolymer micelles with hydrophobic core and hydrophilic shell. The micellar system exhibited dual responsive behavior with respect to light and temperature. Although not novel but such systems are not common. Authors have made an attempt to study the release of cargo (dyes in this case) from the micellar synthesized system. Overall, the paper has sufficient characterizations supporting the claims in the manuscript. Having said that, if accepted, I believe the paper will be a good addition to the field of stimuli-responsive micellar system. I am listing some of the concerns that I feel needs addressing before consideration for publication.

  1. Calculate and present the loading capacity (LC)/loading efficiency of the micelles. This will provide a direct comparison with other materials with similar application.
  2. Authors have studied Nile red as a cargo material. What about other hydrophobic/hydrophilic material? What is the loading and release capacity of such materials for the studied micelles?
  3. Author’s claim of bowl-shaped micelles is not clear from the TEM image in Figure 6 (line 192-193, page 6 of 13). Please provide different angle image where the bowl shape micelles are visible clearly.
  4. Like comment 5, similar analysis should be provided for figure 7.
  5.  From TEM images, there is difference in the micelle’s diameter. Can authors provide and average values with standard deviation in diameter of the micelles from image analysis?
  6. Toxicity of micelles is one of the major challenge for drug delivery. Authors should add a comment on this in the manuscript.
  7. How about stability of these micelles under physiological conditions for sufficiently long periods, especially considering presence of diazo bond in the micelles?
  8. What is the effect on micelles upon synergistic exposure to the stimuli i.e. simultaneous exposure too heat and light and its effect on cargo (dye) release?
  9. Authors mention use of this micellar system for sensors, did they observe hysteresis upon repeated exposure to stimuli (heat and light)? If yes, what is the hysteresis value? If no, what is the reason for such highly responsive system even though the micelles are made from random polymerization and not using controlled synthesis techniques like ATRP or RAFT?

       Other comments:

  1. Line 76 in introduction “The hydrodynamic radius of the micelles occurred a significant change with the ambient temperature” doesn’t sound grammatically correct. Please re-write this line.
  2. Arrange Scheme 1 along with the text in the text of the manuscript.
  3. Line 235-236 does not sound correct grammatically correct “It is 50 min that the absorption band was not change, indicating the photoisomerisation has gotten to the equilibrium of cis- to trans-“.
  4. For all the graphs, the font size can be increased. Current font size makes it difficult to read.

Reviewer 2 Report

Yan et al. synthesised PMAAAB-ran-PHEA random block copolymer and prepared micelles of it. They observed thermo and photo responsive nature of these micelles. I have following minor comments on the manuscript:

1) Inconsistent use of Rh. At some places, it is hydrodynamics radius (Figure 5 and Table 2) while at few other places it is hydrodynamic diameter (Figure 7)

2) Authors were able to find LCST temperature and the size of the free copolymer chains at that temperature. It will be good to calculate and report the aggregation number of the micelles too at different temperatures.

3) Authors have observed increase in size of micelles for NR-S3 as compared to S3. Several such observations have been made in the literature for different types of micelles. Appropriate citations of such literature will be helpful

Round 2

Reviewer 1 Report

  1. The focus of this paper is to study the photo- and thermo-responsiveness of random

copolymer micelles with Nile red as a fluorescent probe, so as to lay the foundation for

possible applications. Therefore, the loading capacity/loading efficiency of micelles is not

examined. This should be the direction of our next research.

I am still critical on the views of the authors of this manuscript on loading capacity/loading efficiency calculations. This calculation is merely theoretical (based on concentration curves of Nile red which authors have already presented in the paper). I agree that the current manuscript is the foundational work for possible applications ahead, but current incomplete investigation of material properties would not warrant future studies. Crucial basic studies can form solid foundation for future application.

  1. According to the research, we found that Nile red with fluorescence property has a lot of

help for studying the micelles with hydrophobic/hydrophilic changes or polarity changes

under external stimuli, so we think that can use Nile red as fluorescent probes to study a

variety of responsiveness for the micelles occurring hydrophobic/hydrophilic changes or

polarity changes under external stimuli. In this research, we did not investigate the loading

and release capacity of hydrophobic substances, so we could not give a definite answer.

This reviewer agrees partially with the authors viewpoint but it again brings me to my first question on LC/LE.

  1. In Figure 6, it is difficult to see that the morphology of S1 and S2 micelles is bowl

shaped, so it has been deleted in the revised manuscript, and the TEM map of S3 has been

replaced. The TEM images of S3, S4 and S5 micelles in the paper are a little too small to

show the bowl shape, but they can be seen when the picture is enlarged. An enlarged TEM is

shown below.

Thanks and noted.

  1. Bowl-shaped micelles formed by amphiphilic random copolymers have been

reported(References[16]: Liu, X.; Kim, J. S.; Wu, J.; Eisenberg, A. Bowl-shaped aggregates

from the self-assembly of an amphiphilic random copolymer of poly(styrene-co-methacrylic

acid). Macromolecules. 2005, 38, 6749-6751.). In the literature, bowl-shaped micelles were

formed when water was gradually added and exceeded the critical water content(CWC%) of

organic solution of random copolymer. Therefore, the formation of porous or bowl-shaped

micelles by PMAAAB-ran-PHEA amphiphilic random copolymer may be related to the

disordered arrangement of hydrophilic and hydrophobic structural units of random

copolymer and the preparation of micelles by dialysis method. We have included the

explanation in our paper and it is shown in red.

Thanks and noted.

  1. According to TEM images, we calculated the average diameter and standard deviation

of S3, S4 and S5 micelles and the results are as follows:

Average values Standard deviation

S3 185 123.0

S4 164 21.9

S5 179 27.1

Thanks and noted.

  1. Based on the light and temperature responsiveness of PMAAAB-ran-PHEA amphiphilic

random copolymer, we suggest that the copolymer micelles have potential application value

in drug delivery. However, it is necessary to determine the cytotoxicity of the micelles for

drug delivery system. According to literature reports, azobenzene compounds have low

toxicity and carcinogenic potential, and there are few reports on the research of drug carriers

based on azobenzene polymers.

Please cite appropriate literature reports on the claim of azobenzene containing polymers as drug carriers (in case authors have not done it). I would recommend that authors should mentions the low cytotoxicity point in the main text and then rationalize the use of this polymer for their studies.

  1. The focus of this paper is to use Nile red as a fluorescent probe to study the photo- and

thermo-responsiveness of random copolymer micelles, so as to lay a foundation for possible

applications. Therefore, the stability of the random copolymer micelles under physiological

conditions is not investigated. As far as we know, azobenzene compounds can be reduced to

hydrogenated azobenzene under basic conditions.

I agree that responsive polymers based on azobenzene are appropriate for the kind of studies presented in this paper but I strongly suggest authors to mention the possible degradation mechanism of azobenzene under physiological conditions. In the response authors claim formation of hydrogenated azobenzene under basic conditions, please mention this in main text and cite appropriate papers.

  1. The release behavior of Nile red during simultaneous exposure to heat and light was not

examined in this study. According to the results of this study, heating can cause the release of

Nile red from the micelles, but the change of polarity within the micelles only occurs under

light, and Nile red is not released. Therefore, we guess that Nile red is released mainly due to

heat when heat and light act simultaneously.

Thanks and noted. Please cite the paper for this claim.

  1. Figure 11 in revised manuscript shows the fluorescence intensity measured after

cis-trans isomerism reaches equilibrium under UV or natural light, so there is no hysteresis

problem. In addition, r1(MAAAB)=0.6882, r2(THP-HEA)=0.6649 were calculated before

the preparation of random copolymer micelles, i.e. r1 and r2 were both less than 1. Therefore,

the compositions of copolymers obtained by conventional radical polymerization were not

significantly different from those obtained by living radical polymerization. If r1 and r2 are

both greater than 1, then the compositions of copolymers obtained by the two polymerization

methods are quite different, and the responsive behaviors of micelles obtained may be

different.

Thanks and noted this a good observation  and I suggest authors to mention this in the main text, maybe under discussion of Figure 7 or 13.

Other comments:

  1. We have revised the statement to “The hydrodynamic radius of the micelles changed

obviously with ambient temperature”.

Thanks and noted.

  1. We have adjusted Scheme1 and the text section appropriately. Thanks and noted.

Author Response

Thank you for your kindly comments for our manuscript (Materials-1483699). On the basis of the first revision, we supplemented the reference [28], and further revised the text description and format, which were marked in the second revised manuscript.

With regard to referee comments, our reply is as follows:

  1. In the first revision, we have revised “hydrodynamic diameter” to “size”. In Figure 7, we revised “size” to “Rh” again.
  2. The micelles formed in this study are porous or bowl-shaped micelles, so the calculation is not carried out. To calculate the aggregation number of the micelles at different temperatures, it is necessary to use the static light scattering (SLS) to determine the weight-average mass(Mw,mic) of the micelle particles, and then use the following formula to calculate the aggregation number (Nagg) of micelles

Nagg=Mw,mic/Mw,m

Where Mw,mic and Mw,m are the weight-average mass of the micelle particles and the polymeric molecules, respectively. Since we have not determined the weight-average mass of micelle particles, we cannot obtain the aggregation number at present.

Here we cited a paper ([28] Jiang W.; Guo J.; Wen W.; Jia Y.-G.; Liu S. Nano-carriers based on pH-sensitive star-shaped copolymers for drug-controlled release. Materials 2019, 12, 1610-1621)
